# Perception of the threat, mental health burden, and healthcare-seeking behavior change among psoriasis patients during the COVID-19 pandemic

Hsien-Yi Chiu[ID][1,2,3,4], Nien-Feng Chang Liao[5], Yu Lin Jr[6], Yu-Huei Huang[ID][7,8]*

1 Department of Medical Research, National Taiwan University Hospital Hsin-Chu Branch, Hsinchu, Taiwan, 2 Department of Dermatology, National Taiwan University Hospital Hsin-Chu Branch, Hsinchu, Taiwan, 3 Department of Dermatology, National Taiwan University Hospital, Taipei, Taiwan, 4 Department of Dermatology, College of Medicine, National Taiwan University, Taipei, Taiwan, 5 Department of Dermatology, China Medical University Hospital, Taichung, Taiwan, 6 Research Services Center for Health Information, Chang Gung University, Taoyuan, Taiwan, 7 Department of Dermatology, Chang Gung Memorial Hospital, Linkou Branch, Taoyuan, Taiwan, 8 School of Medicine, College of Medicine, Chang Gung University, Taoyuan, Taiwan

* huangyh@adm.cgmh.org.tw

## Abstract

This study aimed to investigate the perceived threat, mental health outcomes, behavior changes, and associated predictors among psoriasis patients during the COVID-19 pandemic. The COVID-19 has been known to increase the health risks of patients with psoriasis owing to patients' immune dysregulation, comorbidities, and immunosuppressive drug use. A total of 423 psoriasis patients not infected with COVID-19 was recruited from the Department of Dermatology, National Taiwan University Hospital Hsin-Chu Branch, Chang Gung Memorial Hospital, and China Medical University Hospital from May 2020 to July 2020. A self-administered questionnaire was used to evaluate the perceived threat, mental health, and psychological impact on psoriasis patients using the Perceived COVID-19-Related Risk Scale score for Psoriasis (PCRSP), depression, anxiety, insomnia, and stress-associated symptoms (DAISS) scales, and Impact of Event Scale-Revised (IES-R), respectively. Over 94% of 423 patients with psoriasis perceived threat to be $\geq 1$ due to COVID-19; 18% of the patients experienced psychological symptoms more frequently $\geq 1$, and 22% perceived psychological impact during the pandemic to be $\geq 1$. Multivariable linear regression showed that the higher psoriasis severity and comorbidities were significantly associated with higher PCRSP, DAISS, and IES-R scores. The requirement for a prolonged prescription and canceling or deferring clinic visits for psoriasis treatment among patients are the two most common healthcare-seeking behavior changes during the COVID-19 pandemic. Psoriasis patients who perceived a higher COVID-19 threat were more likely to require a prolonged prescription and have their clinic visits canceled or deferred. Surveillance of the psychological consequences in psoriasis patients due to COVID-19 must be implemented to avoid psychological consequences and inappropriate treatment delays or withdrawal.

**Data Availability Statement:** All relevant data are within the paper and its Supporting Information files.

**Funding:** This work was funded in part by grants from the National Taiwan University Hospital, Hsin-Chu branch (110-HCH045; https://www.hch.gov.tw/) (received by Hsien-Yi Chiu), Chang Gung Memorial Hospital (CMRPG1E0061, CMRPG1E0062, CMRPG1F0161, CMRPG1G0121; https://www.cgmh.org.tw/tw/Services/DeptList/3) (received by Yu-Huei Huang), and Ministry of Science and Technology, Taiwan (MOST 110-2314-B-002-191; https://www.most.gov.tw/) (received by Hsien-Yi Chiu). The funders had no role in study design, data collection and analysis, interpretation of findings, manuscript writing, and target journal selection.

**Competing interests:** All authors have completed the ICMJE uniform disclosure form available at www.icmje.org/coi_disclosure.pdf and declare the following: H.Y.C. received speaking fees from AbbVie, Novartis Pharmaceuticals Corporation, Janssen-Cilag Pharmaceutica, Eli-Lilly, Kyowa Hakko Kirin Taiwan, and Pfizer Limited and conducted clinical trials for Eli-Lilly and Sanofi Pharmaceuticals. Y.H.H. has conducted clinical trials for serving as a principal investigator for Galderma, Eli-Lilly, Novartis Pharmaceuticals Corporation, and Janssen-Cilag Pharmaceutica; received honoraria for serving as an advisory board member for Pfizer Limited, AbbVie, and Celgene; and received speaking fees from AbbVie, Eli-Lilly, and Novartis Pharmaceuticals Corporation. N.F.C.L. have received speaking fees from AbbVie, Novartis, Eli-Lilly Pharmaceuticals Corporation, Janssen-Cilag Pharmaceutica, and Leo Pharm. This does not alter our adherence to PLOS ONE policies on sharing data and materials.

# Introduction

In December 2019, a novel outgoing coronavirus (COVID-19) emerged from Wuhan, Hubei Province, which caused unexplained pneumonia, rapidly spread to over 150 countries globally and became a global public health threat. On September 5, 2021, World Health Organization declared that the pandemic led to 218,946,836 confirmed cases and 4,539,723 deaths worldwide, with a case mortality rate of approximately 2% to 5% [1, 2].

In dermatology, treatment of inflammatory and autoimmune skin diseases commonly involves immunosuppressant use, which may alter the competence of the host immune-surveillance system to combat the virus and has the potential to increase susceptibility, persistence, and reactivation of viral infections [3–6]. It was estimated that 17% of patients with moderate to severe psoriasis required treatments with systemic immunomodulators (such as methotrexate and cyclosporine) or biologics agents (such as tumor necrosis factor (TNF)-α, and interleukin (IL)-17 inhibitors) [7]. Accumulating evidence shows that TNF-α and IL-17 play crucial roles in antiviral immune responses associated with COVID-19 [8, 9]. Patients with psoriasis face a higher risk of serious infections that can lead to hospitalization and increase significant morbidity and/or mortality [10]. The infections can also cause exacerbation of psoriasis [11]. Psoriasis is a T-cell-mediated and chronic disease, associated with risks of comorbidities [12]. An existing study also indicated that people with underlying chronic morbidities were at a greater risk of developing severe symptoms and having poorer clinical outcomes when contracting COVID-19 [13]. These underlying factors render patients with psoriasis vulnerable to the impact of COVID-19 [14, 15].

The uncertainty and consequences associated with morbidity inherent with pandemic outbreaks could induce profound psychosocial impacts on non-infected people, including general populations and patients with chronic health conditions, in addition to the physical illness directly caused by the infection [16–19]. Thus, COVID-19 poses several new concerns and challenges for healthcare professionals when caring for patients with pre-existing immune disorders [20, 21]. During the height COVID-19 pandemic, many psychiatric services were closed, and the lockdown further aggravated patients' vulnerability to mental disorders [22]. A recent study observed that the COVID-19 pandemic had a moderate or worse psychological impact on 20% of patients with rheumatoid arthritis, systemic lupus erythematosus as well as those immunosuppressed and those taking immunosuppressant drugs [23]. A thorough search of academic databases reveals no existing study on the impact of COVID-19 pandemics on psoriasis patients who are free from COVID-19 infections. The present study aims to investigate the perception of the threat, mental health burden, and healthcare-seeking behavior changes among psoriasis patients not infected with the COVID-19 pandemic.

# Methods

This study used a self-administered questionnaire to assess the impact and behavior changes of psoriasis patients during the COVID-19 pandemic. Psoriasis patients not infected with COVID-19 were recruited consecutively from the Department of Dermatology, National Taiwan University Hospital Hsin-Chu branch, Chang Gung Memorial Hospital, and China Medical University Hospital from May 2020 to July 2020. The anamnesis data provide information about the absence of COVID-19 infection in participants. The structured questionnaires are designed to collect patients' demographic data, the perceived threat of COVID-19, the impact of the COVID-19 pandemic on mental health, sleep quality, and behavior changes. The study protocol was reviewed and approved by the institutional review board of National Taiwan University Hospital, Hsin-Chu branch (109-030-E), Chang Gung Medical Foundation, Taiwan (202000851B0), and China Medical University Hospital (CMUH109-REC3-066).

Patients' perceptions of the COVID-19 threat were measured using 13 items (S1 Table) based on a five-point Likert rating scale ranging from 1 (not worried at all) to 5 (very worried). The responses on these items were summed to produce the Perceived COVID-19-Related Risk Scale score for Psoriasis (PCRSP), which ranged from 13 to 85. In addition, participants were asked to choose the top three items of their most concern. Regarding mental health, patients were asked whether they experienced more depression, anxiety, insomnia, and stress-associated symptoms (DAISS) after the COVID-19 than before the outbreak on a five-point scale (from "1 = strongly disagree" to "5 = strongly agree") (S1 Table). Moreover, the psychological impact of COVID-19 was measured using the 22-item Impact of Event Scale-Revised (IES-R; range, 0–88), which was well-validated in the Chinese population to determine the extent of psychological distress after exposure to a public health crisis, such as COVID-19 [24, 25]. The changes in healthcare-seeking behavior and therapy adherence during the COVID-19 pandemic were assessed using seven questions. In S1 Table, these questions inquired whether patients had adopted the prespecified behaviors more frequently at the time of the survey using a five-point Likert rating scale ranging from 1 (strongly disagreed) to 5 (strongly agree) as compared with behaviors adopted in the pre-COVID-19 period. Items in these questionnaires except the IES-R were chosen and modified based on the following: a review of the published literature investigating the impact of COVID-19 [26], Severe Acute Respiratory Syndrome (SARS) [16], Middle East Respiratory Syndrome [27], avian influenza pandemics [28], and consultation with patients and clinical experts. Information about COVID-19-associated quarantined/isolation and psoriasis disease was collected using a questionnaire and/or chart review.

## Statistical analysis

The content validity of PCRSP and DAISS was assessed by seven experts who rated the relevance of each question/item in the questionnaire on a scale of 1 to 4, and the results were presented as the Item Content Validity Index (I-CVI), as the average I-CVIs for all the items in the questionnaire (S-CVI/Ave), and total agreement [29, 30]. Internal consistency and the test-retest reliability of the PCRSP and DAISS were evaluated using Cronbach's coefficient alpha ($\alpha$ > 0.70 is considered to be a good internal consistency) [31]. The intraclass correlation coefficients (ICC) are widely used for the reliability index. ICC $\geq$ 0.6 is considered good reliability [32]. We also evaluated the correlations between these scales using Spearman's correlation coefficients (strong: rho $\geq$ 0.5, moderate: 0.30 < rho < 0.49, weak: 0.10 < rho < 0.29) [33] to ensure convergent validity. Comparisons between two or more groups were conducted using the Mann–Whitney U test and the Kruskal–Wallis test for continuous variables and the chi-squared test or Fisher's exact test for discrete variables. In subgroup analysis, patients were groups using the baseline psoriasis area and severity index (PASI) score, as follows: < 12, 12 $\leq$ PASI < 20, or PASI $\geq$ 20 [34]. Multivariable linear regression models were used to determine potential risk factors for PCRSP, DAISS, and IES-R. Independent variables included age, sex, body weight, the severity of psoriasis, family history of psoriasis, disease duration, comorbidities (including cardiovascular disease, diabetes, hyperlipidemia, and psychiatric disorder), psoriatic arthritis, friends or family member with quarantine experience, and treatment categories (biologics for cutaneous psoriasis, biologics for psoriatic arthritis, traditional antipsoriatic drugs or phototherapy and/or topical drugs only). The R software was used to estimate the target sample size. Since little preliminary data were used in this issue, we determined the sample size based on a previous study [35] that reported gender differences using the IES-R scale during the COVID-19 pandemic. We assumed that the mean (± SD) of the IES-R score for men was 8.56 (±11.86) and that of women was 14.11 (±14.09). For a male-to-female ratio of 3:1, a sample size of 420 patients is required to achieve a 95% power and 5% type 1 error.

Statistical analysis except the sample size estimation was performed using the SPSS 21 statistical package (IBM Corporation, Armonk, NY, USA). A p-value $< 0.05$ (two-tailed) was considered to be statistically significant.

## Results

### Validation of PCRSP and DAISS

The scores for S-CVI/Ave for PCRSP and DAISS were 0.98 and 1.00, respectively. The total agreement was 92.3% and 100%, respectively, which suggested the validity of the questionnaire content [36]. The construct validity of these scales was supported by a strong correlation between DAISS and IES-R (rho = 0.664, $p < 0.001$) and a moderate correlation between PCRCP and DAISS (rho = 0.416, $p < 0.001$) as well as PCRCP and IES-R (rho = 0.435, $p < 0.001$). For reliability, the internal consistency of the PCRCP and DAISS scale, measured by Cronbach's alpha coefficient, was 0.938 and 0.908, respectively. Both PCRCP (ICC = 0.907) and DAISS (ICC = 0.944) showed a good test-retest reliability.

### Perception of COVID-19-related risk

Four hundred twenty-three psoriatic patients not infected with COVID-19 completed the survey. Table 1 shows the demographic and background characteristics of participants. Three

**Table 1. Participant demographic and disease characteristics.**

| Cohort characteristic | N (%) |
|---|---|
| Number of cases | 423 |
| Age (years), median (IQR) | 45.0 (21) |
| Sex (male/female) | 306 (72.3%)/117 (27.7%) |
| Bodyweight (kg), median (IQR) | 76.0 (22) |
| Smoking (Yes/No/Quit) (%) | 126 (29.7%)/228 (53.9%)/69 (16.3%) |
| Alcohol consumption (%) | 78 (18.4%) |
| Family history of psoriasis, % | 27.9% |
| Duration of psoriasis (years), median (IQR) | 13 (16) |
| Hypertension | 27.7% |
| Diabetes | 13.7% |
| Hyperlipidemia | 10.9% |
| Cerebrovascular accident | 0.9% |
| Psoriatic arthritis (%) | 44.0% |
| Severity of psoriasis at baseline | 15.8 ± 8.8 |
| PASI $\geq$ 20 | 78 (18.4%) |
| 12 $\leq$ PASI < 20 | 95 (22.5%) |
| PASI < 12 | 250 (59.1%) |
| Treatment | |
| Biologics for cutaneous psoriasis | 45.1% |
| Biologics for psoriatic arthritis | 19.4% |
| Methotrexate | 25.8% |
| Cyclosporine | 4% |
| Acitretin | 7.1% |
| Phototherapy | 15.1% |
| Topical drugs | 69.3% |

IQR, interquartile range; PASI, Psoriasis Area and Severity Index; IQR, interquartile range.

hundred ninety-nine (94.3%) patients with psoriasis perceived at least one threat from COVID-19 (item score ≥ 3). The top three items that concerned patients the most were the risk of COVID-19 transmission to family members if they contracted COVID-19 (44.9% of patients), drug shortages for psoriasis therapy during the COVID-19 pandemic (32.2%), and fear of contracting COVID-19 in hospitals when attending the dermatology clinic for psoriasis (30.9%) (S2 Table). Female patients with psoriasis, who had a comorbidity, had a psoriatic arthritis (PsA), had a psoriasis duration >15 years, had a higher PASI score at baseline (PASI ≥20), and whose friends or a family member had quarantine experience reported higher average PCRSP scores (aPCRSP) compared, respectively, with male patients (p = 0.004), had no comorbidity (p = 0.001), no PsA (p = 0.003), a duration ≤15 (p = 0.010), lower PASI score (PASI ≥20 vs. PASI <12, p = 0.001; PASI ≥20 vs. 12 ≤ PASI < 20, p = 0.011), and without quarantine experience (p = 0.001). Moreover, patients who were treated with biologics reported higher average of aPCRSP score compared with those who were treated with phototherapy and/or topical drugs only (biologics for cutaneous psoriasis vs. phototherapy/topical, p = 0.005; biologics for PsA vs. phototherapy/topical, p = 0.002) (Fig 1).

## Impact of COVID-19 on mental health and healthcare-seeking behavior

A considerable number of participants with psoriasis (17.7%) experienced at least one psychological symptom, more frequently (item score ≥ 4) during the COVID-19 pandemic, which included depression (13.5%), stress (9.7%), insomnia (5.9%), and anxiety (5.4%) than before the outbreak. For IES-R, a total of 263 (21.7%) reported a psychological impact (IES-R ≥ 24) [37] of COVID-19. Psoriasis patients with higher PASI score at baseline of (PASI ≥ 20) and comorbidities scored higher in DAISS and IES-R than those with less severe disease and no comorbidities, respectively (PASI ≥ 20 vs. PASI < 12, p = 0.012 for DAISS and p = 0.002 for IES-R; PASI ≥ 20 vs. 12 ≤ PASI < 20, p = 0.009 for DAISS and p = 0.011 for IES-R) (Fig 2).

Thirty-one percent of survey patients prefer switching to a long-term drug prescription for psoriasis or drugs that can decrease clinic visits for a long term during the COVID-19 pandemic (S3 Table). Compared with patients who had a PCRSP < 3 and disease duration >15 years, those with aPCRSP ≥ 3 (p = 0.045) and disease duration ≤ 15 years (p = 0.005) were more likely to require a prolonged prescription. Thirteen percent of patients postponed, canceled, or reduced scheduled clinic visits for psoriasis treatment, particularly those whose friends or family members had a quarantine experience (p = 0.001) and those with aPCRSP ≥ 3 (p = 0.044) (Fig 3). Fewer than 5 percent of patients interrupted or discontinued with phototherapy, oral drugs, or biologics for psoriasis treatment and did not take medications for psoriasis according to their doctor's instructions or postponed healthcare-seeking behavior for other non-COVID-19 diseases owing to risks associated with COVID-19.

## Factors influencing the impact of COVID-19 on psoriasis patients

Multivariable linear regression showed that more severe psoriasis measured by PASI, comorbidities, female sex, duration of psoriasis >15 years, and friends or a family member with quarantine experience was significantly associated with a higher aPCRSP. Similarly, a higher PASI, comorbidity, and female sex were significantly associated with DAISS. A higher PASI and comorbidities were associated with a higher IES-R score (Table 2).

## Discussion

The COVID-19 pandemic has brought the risks of physical illness or death from viral infection and unbearable psychological pressure [26]. The outcomes of a survey that evaluated the severity of the COVID-19 outbreak in China showed that approximately one-third of the

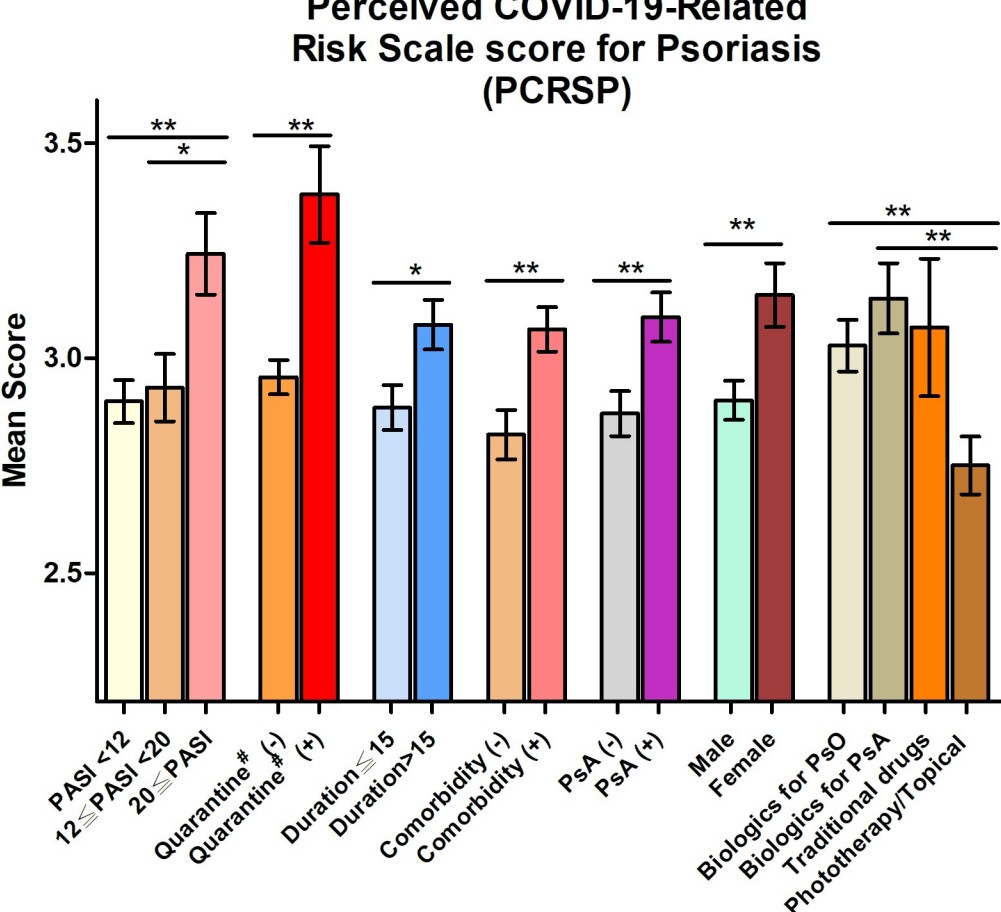

**Fig 1. Perceived COVID-19-Related Risk Scale score for Psoriasis.** Perceived COVID-19-Related Risk Scale score for Psoriasis stratified by demographic and psoriasis disease characteristics. PASI, Psoriasis Area and Severity Index, PsA, psoriatic arthritis, PsO, cutaneous psoriasis. Bars for each estimate indicate the standard error of the mean. #Friends or a family member with or without quarantine experience. *P < 0.05; ** P < 0.01, ***P < 0.001. Comparisons between groups were performed using the Mann–Whitney U test except for the subgroups that were stratified by PASI and treatment modalities, assessed using the Kruskal–Wallis test.

participants reported moderate-to-severe anxiety, while more than half rated the psychological impact as moderate-to-severe [26]. Psoriasis is a systemic inflammatory disease characterized by immune dysregulation, and it is independently linked to a risk of serious infection [38]. Patients with psoriasis are more likely to face risks of comorbidities (such as hypertension, diabetes, and cardiovascular disease) [39], associated with high COVID-19 fatality [13]. A significant body of research also indicates that type I interferon, TNF-$\alpha$, B-cell released antibodies, and other cytokines play a significant role in the viral immune response that combats infection against viral pathogens and promotes clearance [40]. Considering the medication's mechanism, psoriatic patients, taking TNF-$\alpha$ inhibitors, abatacept (CTLA-4 inhibitor), and ustekinumab (IL-12/23 inhibitor), known to modulate and blunt Th1 responses [39], are concerned about the possibility of increased susceptibility to COVID-19 infection. Therefore, facing this large-scale infectious outbreak, patients with psoriasis are theoretically vulnerable to the psychological impact caused by COVID-19. To support this notion, our results showed that 94.3% of patients with psoriasis perceived at least one threat caused by COVID-19, and 88.6% of patients with psoriasis perceived at least two threats caused by COVID-19.

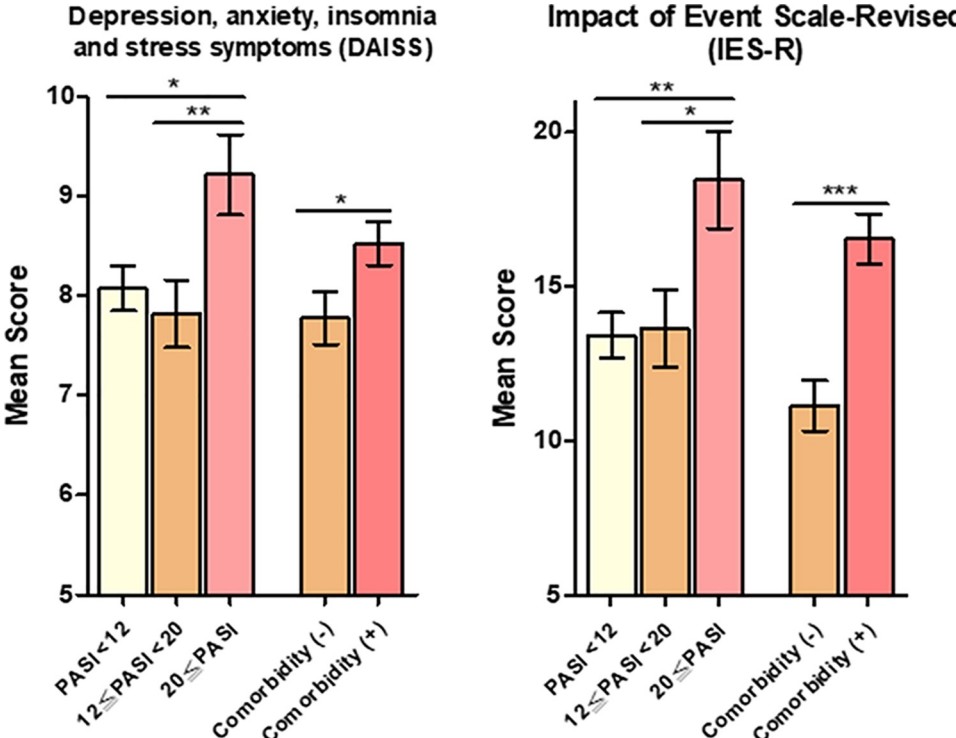

**Fig 2. Psychological impact of the COVID-19 pandemic.** Comparison of scores from the IES-R and a scale measuring DAISS between psoriasis patients who were stratified by psoriasis severity and comorbidity. PASI, Psoriasis Area, and Severity Index. $^*P < 0.05$; $^{**} P < 0.01$, $^{***}P < 0.001$. Comparisons between groups stratified by PASI and comorbidity were performed using the Kruskal–Wallis test and the Mann–Whitney U test, respectively.

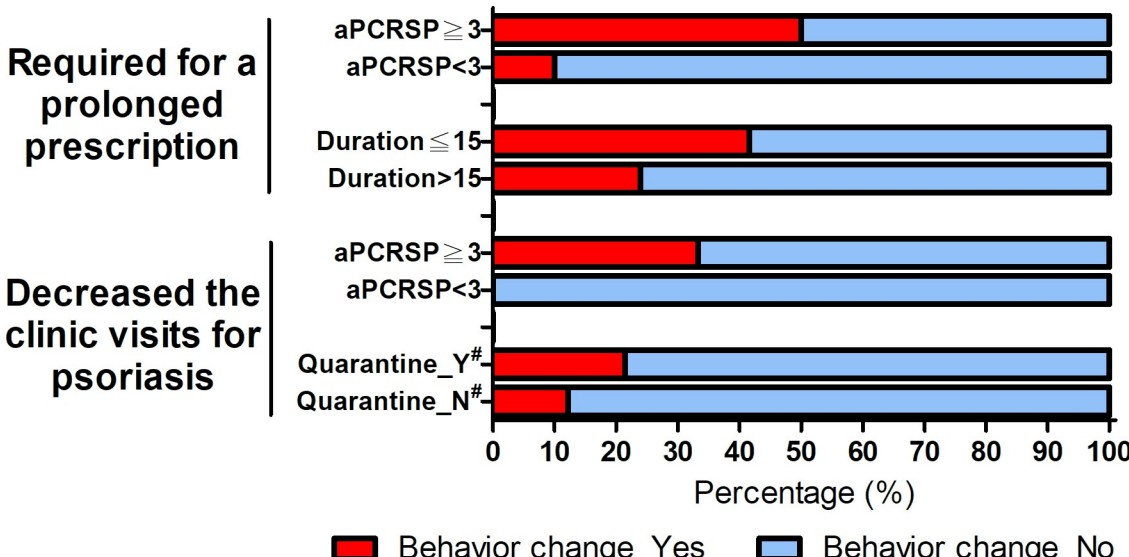

**Fig 3. Healthcare-seeking behavior changes.** Changes in healthcare-seeking behaviors in psoriasis patients during the COVID-19 pandemic among psoriasis patients stratified by perceived COVID-19-related risk (average PCRSP score, aPCRSP), duration of psoriasis, and friends or a family member with and without quarantine experience. * Quarantine_Y, Friends or a family member with quarantine experience, Quarantine_N, Friends or a family member without quarantine experience.

**Table 2. Results of multiple linear regression on factors associated with PCRSP, DAISS, and IES-R.**

| | PCRSP[†] | | | DAISS | | | IES-R | | |
|---|---|---|---|---|---|---|---|---|---|
| **Variables** | **β** | **p-value** | **95% CI** | **β** | **p-value** | **95% CI** | **β** | **P-value** | **95% CI** |
| Sex | 0.129 | 0.008** | 0.06–0.40 | 0.101 | 0.046* | 0.01–1.57 | 0.008 | 0.868 | -2.48–2.94 |
| Severity of psoriasis[†] | 0.142 | 0.005** | 0.04–0.25 | 0.105 | 0.045* | 0.01–0.93 | 0.150 | 0.004** | 0.77–3.97 |
| Disease duration[§] | 0.100 | 0.039* | 0.01–0.32 | -0.009 | 0.867 | -0.76–0.64 | 0.002 | 0.975 | -2.41–2.49 |
| Comorbidity | 0.172 | <0.001*** | 0.13–0.44 | 0.125 | 0.014* | 0.18–1.61 | 0.232 | < 0.001*** | 3.40–8.34 |
| Psoriatic arthritis | 0.076 | 0.123 | -0.03–0.28 | 0.085 | 0.099 | -0.11–1.31 | 0.035 | 0.485 | -1.60–3.36 |
| Friends or a family member had quarantine experience | 0.117 | 0.016* | 0.10–0.95 | 0.006 | 0.910 | -1.82–2.04 | 0.053 | 0.284 | -3.05–10.4 |

DAISS, depression, anxiety, insomnia and stress-associated symptoms, IES-R, Impact of Event Scale-Revised, PCRSP, Perceived COVID-19-Related Risk Scale score for Psoriasis.

*p < 0.05

**p < 0.01

***p < 0.001.

† Average PCRCP score; † Divided into Psoriasis Area and Severity Index (PASI) ≥20, 12 ≤ PASI < 20, and PASI <12 groups

§ Divided into duration >15 and ≤15 years groups.

Moreover, psoriasis severity and duration, PsA, and comorbidities are vulnerability factors associated with patients' perception of the COVID-19 threat, as shown by PCRSP. Patients with more severe psoriasis and comorbidity also reported more psychological symptoms caused by COVID-19 as revealed by DAISS and IES-R scales. The reasons for these associations are not fully understood. However, a recent meta-analysis shows that chronic or severe physical illness is associated with an increased risk of mental disorders, which include anxiety disorder, depression, bipolar disorders, and schizophrenia [41]. Thus, it is reasonable to argue that the accumulated stress or burden associated with a serious underlying physical illness or comorbidities may intensify the perception of danger and increase an individual's lifetime vulnerability to mental disorder. A recent study found that higher levels of depression and anxiety are found in patients with severe psoriasis [42, 43]. Moreover, there is a significant correlation between psoriasis severity, perceived stress, and mood alterations [44, 45]. Patients with PsA also face a high risk of mental disorders, such as depression, which appears to be greater than patients with cutaneous psoriasis only [46].

Similar to the psychological burden caused by SARS [47, 48], our regression analysis indicated that the female sex was an independent factor associated with a higher likelihood of reporting depression, anxiety, insomnia, and distress from the threat of COVID-19, as shown by PCRSP [49]. A growing body of evidence indicates that women are more vulnerable to most traumatic and disaster events and they face an increased risk of developing psychopathological consequences, such as psychiatric disorders [50–52].

Compared with the percentage of patients who perceived a COVID-19 threat and psychological impact, the percentage of change in healthcare-seeking behaviors among psoriasis patients was less prominent in the present study. This was probably because our government initiated a rapid and effective strategy to combat the COVID-19 outbreak, using real-time surveillance for case identification, border control, and quarantine. The government also allocated anti-pandemic resources [53, 54] to mitigate the COVID-19 impact, prevent the breakdown of the medical care system, and maintain easy and safe access to health care. Our results revealed that a greater perception of a COVID-19-related threat was significantly associated with changes in healthcare-seeking behaviors. A higher proportion of canceling or deferring clinic visits by psoriasis patients was recorded by those who had friends or a family member with quarantine experience because of the fear of exposure to COVID-19. Previous

studies also found that higher perceived risk, threat, or danger was related to protective or preventive behavior changes during the COVID-19 outbreak [55, 56]. The inverse association between the disease duration and a prolonged prescription was probable because patients with a longer disease duration were more likely to believe the benefits of regular assessment and treatment for psoriasis outweighed the potential risks caused by COVID-19.

This study has some limitations. First, the Taiwanese population is the selected patient for clinical research, and it might be difficult to generalize the results to other ethnic groups in other regions. Second, information was collected quickly over a relatively few weeks because of the uncertainty associated with the disease under investigation. Thus, the possibility of selection bias may be possible.

In conclusion, we identified a major perception of threat and mental health burden among psoriasis patients during the COVID-19 outbreak. Over 94% of patients perceived at least one threat caused by COVID-19, and 18% of patients experienced $\geq 1$ psychological symptom more frequently, and 22% of patients experienced psychological impact during the pandemic. Patients with more severe psoriasis, comorbidity, and female sex face a higher risk of perceiving a greater threat and are more likely to experience psychological symptoms caused by COVID-19. The requirements for a prolonged prescription and reducing clinic visits for psoriasis treatment were the two most common healthcare-seeking behavior changes during the pandemic, significantly associated with patients' awareness of a threat due to COVID-19. Continuous surveillance of the psychological consequences caused by the COVID-19 pandemic in patients with psoriasis must be immediately implemented along with the provision of mental health support, particularly for vulnerable groups to mitigate the impact of the COVID-19 pandemic.

## Supporting information

**S1 Data.**
(XLS)

**S1 Table. Questionnaire.**
(DOCX)

**S2 Table. Perception of the COVID-19 threat among patients with psoriasis.**
(DOCX)

**S3 Table. Psychological impact and behavioral changes caused by the COVID-19 pandemic on patients with psoriasis.**
(DOCX)

## Author Contributions

**Conceptualization:** Hsien-Yi Chiu, Nien-Feng Chang Liao, Yu Lin Jr, Yu-Huei Huang.

**Data curation:** Hsien-Yi Chiu, Nien-Feng Chang Liao, Yu Lin Jr, Yu-Huei Huang.

**Formal analysis:** Hsien-Yi Chiu, Nien-Feng Chang Liao, Yu Lin Jr, Yu-Huei Huang.

**Funding acquisition:** Hsien-Yi Chiu, Yu-Huei Huang.

**Investigation:** Hsien-Yi Chiu, Nien-Feng Chang Liao, Yu Lin Jr, Yu-Huei Huang.

**Methodology:** Hsien-Yi Chiu, Nien-Feng Chang Liao, Yu Lin Jr, Yu-Huei Huang.

**Project administration:** Hsien-Yi Chiu, Nien-Feng Chang Liao, Yu Lin Jr, Yu-Huei Huang.

**Resources:** Hsien-Yi Chiu, Nien-Feng Chang Liao, Yu Lin Jr, Yu-Huei Huang.

**Software:** Hsien-Yi Chiu, Yu Lin Jr, Yu-Huei Huang.

**Supervision:** Hsien-Yi Chiu, Yu-Huei Huang.

**Validation:** Hsien-Yi Chiu, Nien-Feng Chang Liao, Yu Lin Jr, Yu-Huei Huang.

**Visualization:** Hsien-Yi Chiu, Nien-Feng Chang Liao, Yu Lin Jr, Yu-Huei Huang.

**Writing – original draft:** Hsien-Yi Chiu, Nien-Feng Chang Liao.

**Writing – review & editing:** Yu Lin Jr, Yu-Huei Huang.

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
