## [Decision Letter · Decision Letter 0]

3 Sep 2021

PONE-D-21-15316

The perception of the threat, mental health burden and healthcare- seeking behavior changes amidst COVID-19 pandemic among psoriasis patients

PLOS ONE

Dear Dr. Huang,

Thank you for submitting your manuscript to PLOS ONE. After careful consideration, we feel that it has merit but does not fully meet PLOS ONE’s publication criteria as it currently stands. Therefore, we invite you to submit a revised version of the manuscript that addresses the points raised during the review process.

We look forward to receiving your revised manuscript.

Kind regards,

Sinan Kardeş, M.D.

Academic Editor

PLOS ONE

3.Thank you for stating the following in the Acknowledgments/Funding Section of your manuscript:

“This work was funded in part by grants from the National Taiwan University Hospital, Hsin-Chu branch (110-HCH045) and Chang Gung Memorial Hospital (CMRPG1E0061, CMRPG1E0062, CMRPG1F0161, CMRPG1G0121). The funders had no role in study design, data collection and analysis, interpretation of findings, manuscript writing, and target journal selection. We note that you have provided funding information that is not currently declared in your Funding Statement. However, funding information should not appear in the Acknowledgments section or other areas of your manuscript. We will only publish funding information present in the Funding Statement section of the online submission form.

“Yes, this work was funded in part by grants from the National Taiwan University Hospital, Hsin-Chu branch (110-HCH045; https://www.hch.gov.tw/) (received by Hsien-Yi Chiu) and Chang Gung Memorial Hospital (CMRPG1E0061, CMRPG1E0062, CMRPG1F0161, CMRPG1G0121; https://www.cgmh.org.tw/tw/Services/DeptList/3) (received by Yu-Huei Huang). The funders had no role in study design, data collection and analysis, interpretation of findings, manuscript writing, and target journal selection.”

“All authors have completed the ICMJE uniform disclosure form available at www.icmje.org/coi_disclosure.pdf and declare the following: H.Y.C. received speaking fees from AbbVie, Novartis Pharmaceuticals Corporation, Janssen-Cilag Pharmaceutica, Eli-Lilly, Kyowa Hakko Kirin Taiwan, and Pfizer Limited and conducted clinical trials for Eli-Lilly and Sanofi Pharmaceuticals. Y.H.H. has conducted clinical trials for serving as a principal investigator for Galderma, Eli-Lilly, Novartis Pharmaceuticals Corporation, and Janssen-Cilag Pharmaceutica; received honoraria for serving as an advisory board member for Pfizer Limited, AbbVie, and Celgene; and received speaking fees from AbbVie, Eli-Lilly, and Novartis Pharmaceuticals Corporation. N.F.C.L. have received speaking fees from AbbVie, Novartis, Eli-Lilly Pharmaceuticals Corporation, Janssen-Cilag Pharmaceutica, and Leo Pharm.”

Additional Editor Comments (if provided):

Reviewers' comments:

Reviewer's Responses to Questions

**Comments to the Author**

1. Is the manuscript technically sound, and do the data support the conclusions?

Reviewer #1: Yes

Reviewer #2: Yes

Reviewer #3: Yes

Reviewer #4: No

Reviewer #5: Yes

2. Has the statistical analysis been performed appropriately and rigorously? 

Reviewer #1: Yes

Reviewer #2: Yes

Reviewer #3: Yes

Reviewer #4: No

Reviewer #5: Yes

3. Have the authors made all data underlying the findings in their manuscript fully available?

Reviewer #1: Yes

Reviewer #2: Yes

Reviewer #3: No

Reviewer #4: No

Reviewer #5: Yes

4. Is the manuscript presented in an intelligible fashion and written in standard English?

Reviewer #1: Yes

Reviewer #2: Yes

Reviewer #3: Yes

Reviewer #4: No

Reviewer #5: Yes

5. Review Comments to the Author

Reviewer #1: The study was well conducted and rather relevant to current issues of the pandemic. However, it seems that psoriasis and COVID did not really impact patients' behavior in a different way from other diseases or even the lay public. Patients still continue their treatments prescribed.

Reviewer #2: The manuscript entitled 'The perception of the threat, mental health burden and healthcare-seeking behaviour changes admist COVID-19 pandemic among psoriatic patients' addressed the question of whether perceived threat, mental health outcomes, behavior chenges, and associated predictors among patients admist COVID-19 pandemic. To do so, the 423 patients suffering from psoriasis were asked by using questionnaires such as PCRSP, DAISS and IES-R, showing the increase in perceived threat in most of the patients and association of a higher psoriatic severity and comorbidity with a higher PCRSP, DAISS and IER-S.

Since most of the patients presented here were Taiwanese, it might be difficult to generalize the conclusions drawn by the authors over the different social conditions. However alternatively , I would think that this kind of work should be done in different regions and/or ethnics, which may contribute to our understanding on mental influence of COVID-19 pandemics depending on the different social conditions. Conclusively, this paper should appear medical practitioners such as medical doctors, nurses and co-medics. In this context, I think that this paper may stand on the general readership.

Reviewer #3: The work is interesting and well prepared, but some information/changes need to be considered.

Methods

It is necessary to state whether the project was approved by a research ethics committee and the approval number

Results

"Four hundred and twenty-three psoriatic patients not infected with COVID-19 completed the survey."

Establish whether the criteria for including patients as "psoriatic patients not infected" were based on anamnesis data or if patients were tested to exclude infection.

References

Some references are not placed within the journal's norms

Reviewer #4: In this study the authors assess the perceived threat, mental health outcomes, behavior changes, and associated predictors among psoriasis patients amidst COVID-19 pandemic. The authors show that higher psoriasis severity and comorbidity of psoriasis is associated with higher PCRSP, DAISS and IES-R. I think it is important to address the effects of COVID 19 pandemic on mental and physical status of those affected with chronic diseases. The current study merely describes that the pandemic had led to an adverse psychological impact correlated with disease severity. I do not see this study to be adding new information specifically for psoriasis affected population.

The manuscript needs major English editing.

Abstract:

Line 102-104 is vague. It would be better to rephrase.

Introduction:

Mere general information was introduced in this section. The authors fail to convey their message on why they find psoriasis different from many other conditions in regards to COVID-19 induce adverse psychological impact. There are many chronic diseases that the concept of this study could be applied for them, the authors need to clarify what they find different about psoriasis.

Methods:

Please provide information on power and sample size calculation

Results/Discussion:

The authors claim that regression showed being female was an independent factor for COVID19 related adverse psychological effect. Psoriasis has a higher female to male ratio and some subtypes have more severity in males, however the current study’s cohort was 72.3% male and 27.3% female. What measures have the authors taken to avoid selection bios?

Discussion needs major English editing

Data is not available in supplementary files. The authors merely provided a questionnaire.

Reviewer #5: Considerations in Introduction

Line 132: “The pandemic, as declared by the World Health Organization on 06 May 2021,…"

The date of declaration of the COVID-19 pandemic by the WHO needs correction.

Lines 139, 140, and 141: Psoriasis, a T-cell mediated disease, it was estimated that 17% of patients with moderate to severe psoriasis required systemic immunomodulators or biologic agents, which keeps rising recently [7-9].

The references provided by the authors do not support this statement.

Lines 143, 144, and 145 “Psoriasis is not only a chronic disease itself but also associated with many comorbidities. These underlying factors render patients with psoriasis vulnerable to impact caused by COVID-19.”

The reference is missing.

Consideration in discussion:

Lines 344, 345 and 346 “These findings are consistent with previous studies showing having a higher perceived risk, threat and danger were related to behavior changes during the outbreak of SARS [49,50]”.

Reference 49 does not converge with the results obtained with this study, thus requiring a correction in this statement.

6. PLOS authors have the option to publish the peer review history of their article (what does this mean?). If published, this will include your full peer review and any attached files.

Reviewer #1: No

Reviewer #2: No

Reviewer #3: No

Reviewer #4: No

Reviewer #5: No

---

## [Author Response · Author response to Decision Letter 0]

8 Oct 2021

Reviewer #1: 

Comment#1: The study was well conducted and rather relevant to current issues of the pandemic. However, it seems that psoriasis and COVID did not really impact patients' behavior in a different way from other diseases or even the lay public. Patients still continue their treatments prescribed.

Response 1: Thank you for the comments. Distinct from patients with other diseases, such as diabetes and hypertension, the immunosuppressive drugs for psoriasis therapy, immune dysregulation, and comorbidities associated with psoriasis all render patients with psoriasis more vulnerable to the impact of COVID-19 than patients with other chronic diseases [1, 2]. As described in the manuscript, our survey found that the COVID-19 pandemic has created substantial challenges to the healthcare-seeking behavior among psoriasis patients. Although only less than 5% of patients discontinued their treatment for psoriasis, a substantial portion (13%) of patients postponed, canceled, or decreased the scheduled clinic visits for their psoriasis. Previous studies have suggested that a delay or disruption of continuity of regular medical care can lead to nonadherence to treatment and increase the morbidity and mortality risk [3-5]. Our study reminds physicians who provide care for psoriasis patients to be aware of the changes in healthcare-seeking behavior and medication prescription patterns, which might mitigate the probability of inappropriate treatment delays, non-adherence, or withdrawal resulting in subsequent worsening of psoriasis. 

Reviewer #2: 

Comment #1: The manuscript entitled 'The perception of the threat, mental health burden and healthcare-seeking behaviour changes admist COVID-19 pandemic among psoriatic patients' addressed the question of whether perceived threat, mental health outcomes, behavior chenges, and associated predictors among patients admist COVID-19 pandemic. To do so, the 423 patients suffering from psoriasis were asked by using questionnaires such as PCRSP, DAISS and IES-R, showing the increase in perceived threat in most of the patients and association of a higher psoriatic severity and comorbidity with a higher PCRSP, DAISS and IER-S. Since most of the patients presented here were Taiwanese, it might be difficult to generalize the conclusions drawn by the authors over the different social conditions. However alternatively, I would think that this kind of work should be done in different regions and/or ethnics, which may contribute to our understanding on mental influence of COVID-19 pandemics depending on the different social conditions. Conclusively, this paper should appear medical practitioners such as medical doctors, nurses and co-medics. In this context, I think that this paper may stand on the general readership.

Response 1: Thank you for the comments. Our study only included the Taiwanese population. It might be difficult to generalize the conclusions over different regions and/or ethnics. The applicability to a broader population requires further research. We have reported this limitation in the revised Discussion section.

Reviewer #3: 

The work is interesting and well prepared, but some information/changes need to be considered.

Comment #1: Methods

It is necessary to state whether the project was approved by a research ethics committee and the approval number

Response 1: Thank you for the constructive comment. We have stated the IRB approval and approval number in the revised manuscript. 

Comment #2: Results

"Four hundred and twenty-three psoriatic patients not infected with COVID-19 completed the survey."

Establish whether the criteria for including patients as "psoriatic patients not infected" were based on anamnesis data or if patients were tested to exclude infection.

Response 2: Thank you for the suggestions. In the revised manuscript, we indicated that the absence of COVID-19 infection in participants was based on anamnesis data. 

Comment #3:References

Some references are not placed within the journal's norms

Response 3: Thank you for the comment. We have rechecked and reformatted the reference lists according to the journal’s requirements. 

Reviewer #4: 

Comment #1: In this study the authors assess the perceived threat, mental health outcomes, behavior changes, and associated predictors among psoriasis patients amidst COVID-19 pandemic. The authors show that higher psoriasis severity and comorbidity of psoriasis is associated with higher PCRSP, DAISS and IES-R. I think it is important to address the effects of COVID 19 pandemic on mental and physical status of those affected with chronic diseases. The current study merely describes that the pandemic had led to an adverse psychological impact correlated with disease severity. I do not see this study to be adding new information specifically for psoriasis affected population.

Response 1: Thank you for the comments. The psychological and behavioral impact of the COVID-19 crisis is highly heterogeneous among different regions and specific populations [6, 7]. Distinct from the general population, patients with psoriasis are generally considered to be a vulnerable population during the COVID-19 pandemic because of their immune system dysfunction, immunosuppressive medication use, and associated comorbidities [1, 2]. However, to date, there is little information about the psychological impact and behavioral changes caused by COVID-19, especially for the psoriasis-affected population. In the present study, we used a questionnaire that was specially designed for patients with psoriasis, which was based on the distinct features of this population, to assess the perceived threat, mental health outcomes, and behavior changes among patients with psoriasis during the COVID-19 pandemic. We validated the questionnaire. Our results described accurate estimates of the mental health burden, degree of concern about the pandemic, and changes in healthcare-seeking behavior. A considerable proportion of participants with psoriasis (17.7%) had experienced at least one psychological symptom, suggesting that mental health interventions are critically required for patients with psoriasis during the COVID-19 pandemic. These findings can be used to plan for mitigating measures by the health authorities. Our study also found factors that were associated with a high perceived threat and psychological symptoms among patients with psoriasis. These findings can help to identify and target high-risk psoriatic patients with a high mental health burden who may need specific psychological intervention or counseling support to prevent adverse mental health outcomes.

Studies have found the COVID-19 pandemic and global efforts to contain its spread (such as lockdown measures and transportation shutdowns) have led to limited access to healthcare, resulting in decreased service delivery and utilization for the general population [8, 9]. However, the absolute magnitude of the COVID-19 impact on healthcare-seeking behavior specifically in patients with psoriasis remains mostly unclear. The present study revealed that health care utilization for psoriasis treatment has changed in a large proportion of psoriasis patents (31%) during the COVID-19 pandemic. Understanding healthcare-seeking behavior changes in people with psoriasis might help clinicians better address their patients’ needs and inform policies and health authorities to protect this potentially vulnerable population.

Comment #2:The manuscript needs major English editing.

Response 2 Thank you for the comment. We have had our manuscript reviewed and edited by a native English-speaking editor (UNIVERSAL LINK CO., LTD.). The certificate of English editing has been provided. 

Comment #3:Abstract:

Line 102-104 is vague. It would be better to rephrase.

Response 3: Thank you for the comment. We have rephrased the abstract as follows:

“The requirement for a prolonged prescription and canceling or deferring clinic visits for psoriasis are the two most common healthcare-seeking behavior changes among patients with psoriasis during the COVID-19 pandemic. Psoriasis patients who perceived a higher COVID-19 threat were more likely to require a prolonged prescription and have their clinic visits cancelled or deferred.” 

Comment #4:

Introduction:

Mere general information was introduced in this section. The authors fail to convey their message on why they find psoriasis different from many other conditions in regards to COVID-19 induce adverse psychological impact. There are many chronic diseases that the concept of this study could be applied for them, the authors need to clarify what they find different about psoriasis.

Response 4: Thank you for the suggestions. We have amended and added several sentences to point out the specific features of psoriasis that are distinct from other chronic diseases. The changes are as follows: 

“It was estimated that 17% of patients with moderate to severe psoriasis required systemic immunomodulators (such as methotrexate and cyclosporine) or biologic agents (such as tumor necrosis factor (TNF)-α and interleukin (IL)-17) inhibitors to treat their psoriasis, which is a T cell-mediated disease [7]. Accumulating evidence has shown that both TNF-α and IL-17 play crucial roles in antiviral immune responses that are associated with COVID-19 [8, 9]. Compared with those without psoriasis, patients with psoriasis had a higher risk of serious infection, leading to hospitalization and significant morbidity and/or mortality [10]. Infections can also cause exacerbation of psoriasis [11]. Moreover, psoriasis is a chronic disease, and it is also associated with many comorbidities [12].” 

Comment #5:Methods:

Please provide information on power and sample size calculation

Response 5:

Thank you for the comments. We had added the description of power and a sample size calculation into the statistical analysis section of the revised manuscript, which is as follows: 

“The target sample size was calculated using R software. Because there were few preliminary data on this issue, we determined the sample size based on a previous study [35] that reported gender differences using the IES-R scale during the COVID-19 pandemic. We assumed that the mean (±SD) IES-R score for men and women was 8.56 (±11.86) and 14.11 (±14.09), respectively. For a male-to-female ratio of 3:1, we needed a sample size of 420 patients for 95% power and 5% type 1 error.” 

Comment #6: Results/Discussion:

The authors claim that regression showed being female was an independent factor for COVID19 related adverse psychological effect. Psoriasis has a higher female to male ratio and some subtypes have more severity in males, however the current study’s cohort was 72.3% male and 27.3% female. What measures have the authors taken to avoid selection bios?

Response 6: Thank you for the comments. Although a slightly higher prevalence of psoriasis was observed in female subjects than male subjects in Germany (0.76% vs. 0.66%), United States (2.5% vs. 1.9%) and Norway (1.6% vs. 1.2%) [11], a nationwide epidemiologic study using Taiwan National Health Insurance (NHI) claims database showed that psoriasis was more frequent in men than in women in Taiwan (61.6% vs. 38.4%) [12]. Thus, in our study, the higher prevalence of psoriasis in men is less likely due to selection bias and suggests that the sex ratio for the prevalence of psoriasis varies between different ethnic groups. 

Comment #7: Discussion needs major English editing

Response 7 Thank you for your comment. We have had our manuscript reviewed and edited by a native English-speaking editor. The certificate of English editing is has been provided. 

Comment #8 Data is not available in supplementary files. The authors merely provided a questionnaire.

Response 8: Thank you for the comments. We have provided additional data in the revised supplementary files. 

Reviewer #5: Considerations in Introduction

Comment #1:Line 132: “The pandemic, as declared by the World Health Organization on 06 May 2021,…"

The date of declaration of the COVID-19 pandemic by the WHO needs correction.

Response 1: Thank you for the constructive comments. We have updated the COVID-19 pandemic statistics and its date of declaration by the WHO. 

Comment #2: Lines 139, 140, and 141: Psoriasis, a T-cell mediated disease, it was estimated that 17% of patients with moderate to severe psoriasis required systemic immunomodulators or biologic agents, which keeps rising recently [7-9].

The references provided by the authors do not support this statement.

Response 2: Thank you for the suggestions. We have revised the reference. According to a questionnaire study that was published by Mrowietz et al. in the British Journal of Dermatology (Br J Dermatol. 2006 Oct;155(4):729-36), 17% of patients with moderate to severe psoriasis required systemic immunomodulators or biologic agents. We have added this reference into the revised manuscript. 

Comment #3: Lines 143, 144, and 145 “Psoriasis is not only a chronic disease itself but also associated with many comorbidities. These underlying factors render patients with psoriasis vulnerable to impact caused by COVID-19.”

The reference is missing.

Response 3: Thank you for the suggestions. We have revised the manuscript and cited the relevant references. 

Comment #4: Consideration in discussion:

Lines 344, 345 and 346 “These findings are consistent with previous studies showing having a higher perceived risk, threat and danger were related to behavior changes during the outbreak of SARS [49,50]”. Reference 49 does not converge with the results obtained with this study, thus requiring a correction in this statement.

Response 4: Thank you for the constructive comments. We have rephrased the sentences and cited new relevant references, which are as follows: 

“Similarly, previous studies also found that a higher perceived risk, threat, or danger was related to protective or preventive behavior changes during the COVID-19 outbreak.” 

References 

1. Strippoli D, Barbagallo T, Prestinari F, Russo G, Fantini F. Biologic agents in psoriasis: our experience during coronavirus infection. Int J Dermatol. 2020;59(8):e266-e7. doi: 10.1111/ijd.15002. PMID: 32516447.

2. J. Liu. Association between Biologic Therapy and COVID-19 Infection Risk in Patients with Psoriasis. Presented at the: American Academy of Dermatology Virtual Meeting Experience 2021 (AAD VMX); Virtual.

3. CDC, National Center for Health Statistics. Excess deaths associated with COVID-19. Atlanta, GA: US Department of Health and Human Services, CDC, National Center for Health Statistics; 2020. https://www.cdc.gov/nchs/nvss/vsrr/covid19/excess_deaths.htm

4. Osendarp S, Akuoku JK, Black RE, Headey D, Ruel M, Scott N, et al. The COVID-19 crisis will exacerbate maternal and child undernutrition and child mortality in low- and middle-income countries. Nature Food. 2021; 2(7):476-84. doi: 10.1038/s43016-021-00319-4.

5. Wang JJ, Levi JR, Edwards HA. Changes in Care Provision During COVID-19 Impact Patient Well-Being. J Patient Exp. 2021;8:23743735211034068. doi: 10.1177/23743735211034068. PMID: 34350341.

6. Ellwardt L, Prag P. Heterogeneous mental health development during the COVID-19 pandemic in the United Kingdom. Sci Rep. 2021; 11(1):15958. doi: 10.1038/s41598-021-95490-w. PMID: 34354201.

7. Glintborg B, Jensen DV, Engel S, Terslev L, Pfeiffer Jensen M, Hendricks O, et al. Self-protection strategies and health behaviour in patients with inflammatory rheumatic diseases during the COVID-19 pandemic: results and predictors in more than 12 000 patients with inflammatory rheumatic diseases followed in the Danish DANBIO registry. RMD Open. 2021; 7(1). doi: 10.1136/rmdopen-2020-001505. PMID: 33402443.

8. Roy CM, Bollman EB, Carson LM, Northrop AJ, Jackson EF, Moresky RT. Assessing the indirect effects of COVID-19 on healthcare delivery, utilization and health outcomes: a scoping review. Eur J Public Health. 2021; 31(3):634-40. doi: 10.1093/eurpub/ckab047. PMID: 33755130.

9. Blumenthal D, Fowler EJ, Abrams M, Collins SR. Covid-19 - Implications for the Health Care System. N Engl J Med. 2020; 383(15):1483-8. doi: 10.1056/NEJMsb2021088. PMID: 32706956.

10. Jo SH, Koo BH, Seo WS, Yun SH, Kim HG. The psychological impact of the coronavirus disease pandemic on hospital workers in Daegu, South Korea. Compr Psychiatry. 2020;103:152213. doi: 10.1016/j.comppsych.2020.152213. PMID: 33096399.

11. Parisi R, Symmons DP, Griffiths CE, Ashcroft DM, Identification, Management of P, et al., Global epidemiology of psoriasis: a systematic review of incidence and prevalence. J Invest Dermatol. 2013;133(2):377-85. doi: 10.1038/jid.2012.339. PMID: 23014338.

12. Tsai TF, Wang TS, Hung ST, Tsai PI, Schenkel B, Zhang M, et al., Epidemiology and comorbidities of psoriasis patients in a national database in Taiwan. J Dermatol Sci. 2011;63(1):40-6. doi: 10.1016/j.jdermsci.2011.03.002. PMID: 21543188.

---

## [Decision Letter · Decision Letter 1]

14 Oct 2021

PONE-D-21-15316R1Perception of the threat, mental health burden, and healthcare- seeking behavior changes among psoriasis patients during the COVID-19 pandemicPLOS ONE

Dear Dr. Huang,

Thank you for submitting your manuscript to PLOS ONE. After careful consideration, we feel that it has merit but does not fully meet PLOS ONE’s publication criteria as it currently stands. Therefore, we invite you to submit a revised version of the manuscript that addresses the points raised during the review process.

We look forward to receiving your revised manuscript.

Kind regards,

Sinan Kardeş, M.D.

Academic Editor

PLOS ONE

Journal Requirements:

Reviewers' comments:

Reviewer's Responses to Questions

**Comments to the Author**

1. If the authors have adequately addressed your comments raised in a previous round of review and you feel that this manuscript is now acceptable for publication, you may indicate that here to bypass the “Comments to the Author” section, enter your conflict of interest statement in the “Confidential to Editor” section, and submit your "Accept" recommendation.

Reviewer #1: All comments have been addressed

Reviewer #2: All comments have been addressed

2. Is the manuscript technically sound, and do the data support the conclusions?

Reviewer #1: (No Response)

Reviewer #2: Yes

3. Has the statistical analysis been performed appropriately and rigorously? 

Reviewer #1: (No Response)

Reviewer #2: Yes

4. Have the authors made all data underlying the findings in their manuscript fully available?

Reviewer #1: (No Response)

Reviewer #2: Yes

5. Is the manuscript presented in an intelligible fashion and written in standard English?

Reviewer #1: (No Response)

Reviewer #2: Yes

6. Review Comments to the Author

Reviewer #1: Although much better, English editing in several places is still necessary.

Just one example in lines 294-5 is provided here: Psoriasis patients with are more prone to develop comorbidities....

Reviewer #2: (No Response)

7. PLOS authors have the option to publish the peer review history of their article (what does this mean?). If published, this will include your full peer review and any attached files.

Reviewer #1: No

Reviewer #2: No

---

## [Author Response · Author response to Decision Letter 1]

26 Oct 2021

Journal Requirements

Please review your reference list to ensure that it is complete and correct. If you have cited papers that have been retracted, please include the rationale for doing so in the manuscript text or remove these references and replace them with relevant current references. Any changes to the reference list should be mentioned in the rebuttal letter that accompanies your revised manuscript. If you need to cite a retracted article, indicate the article’s retracted status in the References list and also include a citation and full reference for the retraction notice.

Response 1:

Thank you for the instructions. We have reviewed our reference list, and no cited papers have been retracted. 

Reviewers’ comments

Reviewer #1: 

Comment#1: Although much better, English editing in several places is still necessary. Just one example in lines 294-5 is provided here: Psoriasis patients with are more prone to develop comorbidities.....

Response 1: Thank you for the comments. We have rephrased this sentence as follows: Patients suffering from psoriasis are more likely to face the risks of comorbidities, such as hypertension, diabetes, and cardiovascular disease. The manuscript has been carefully reviewed by an experienced editor whose first language is English and who specializes in editing papers written by scientists whose native language is not English. The English editing certificate is provided below.

---

## [Editor Report · Decision Letter 2]

28 Oct 2021

Perception of the threat, mental health burden, and healthcare-seeking behavior change among psoriasis patients during the COVID-19 pandemic

PONE-D-21-15316R2

Dear Dr. Huang,

We’re pleased to inform you that your manuscript has been judged scientifically suitable for publication and will be formally accepted for publication once it meets all outstanding technical requirements.

Kind regards,

Sinan Kardeş, M.D.

Academic Editor

PLOS ONE
---

## [Editor Report · Acceptance letter]

1 Dec 2021

PONE-D-21-15316R2 

Perception of the threat, mental health burden, and healthcare-seeking behavior change among psoriasis patients during the COVID-19 pandemic 

Dear Dr. Huang:

I'm pleased to inform you that your manuscript has been deemed suitable for publication in PLOS ONE. Congratulations! Your manuscript is now with our production department. 

Kind regards, 

on behalf of

Dr. Sinan Kardeş 

Academic Editor

PLOS ONE